# Human Immunodeficiency Virus Infected Patients are Not at Higher Risk for Hepatitis E Virus Infection: A Systematic Review and Meta-Analysis

**DOI:** 10.3390/microorganisms7120618

**Published:** 2019-11-26

**Authors:** Pedro Lopez-Lopez, Mario Frias, Angela Camacho, Antonio Rivero, Antonio Rivero-Juarez

**Affiliations:** Infectious Diseases Unit, Instituto Maimónides de Investigación Biomédica de Córdoba (IMIBIC), Hospital Universitario Reina Sofía de Córdoba, Universidad de Córdoba, 14004 Córdoba, Spain; lopezlopezpedro07@gmail.com (P.L.-L.); mariofriascasas@hotmail.com (M.F.); acamachoespejo@gmail.com (A.C.); ariveror@gmail.com (A.R.)

**Keywords:** HEV, HIV, risk factor, systematic review, meta-analysis

## Abstract

Hepatitis E virus (HEV) infection is the most common cause of acute hepatitis in the world. It is not well established whether people infected with the human immunodeficiency virus (HIV) are more susceptible to infection with HEV than people not infected with HIV. Many studies have evaluated this relationship, although none are conclusive. The aim of this systematic review and meta-analysis was to assess whether patients with HIV infection constitute a risk group for HEV infection. A systematic review and meta-analysis was performed in line with Preferred Reporting Items for Systematic Reviews and Meta-Analyses (PRISMA), to find publications comparing HEV seroprevalences among HIV infected and uninfected populations. The analysis was matched by sex, age and geographical area, and compared patients who live with HIV and HIV-negative individuals. The odds ratio (OR) for patients with HIV was 0.87 (95% CI: 0.74–1.03) in the fixed effects meta-analysis and 0.88 (95% CI: 0.70–1.11) in random effects, with I^2^ = 47%. This study did not show that HIV infection was a risk factor for HEV infection when compared with those who are HIV-negative.

## 1. Introduction

Infection with the hepatitis E virus (HEV) is the most common cause of acute hepatitis worldwide [1]. Five major HEV genotypes (1–4 and 7) cause infection in humans [2], in addition, it has been experimentally demonstrated that genotype 8 can infect nonhuman primates, suggesting a potential zoonotic transmission [3]. In most cases, HEV infection is asymptomatic and self-limiting [4], although there are clinical contexts in which the course of acute HEV infection has a worse prognosis, for example, in women who are pregnant and patients with underlying chronic liver disease [5,6]. Furthermore, in immunocompromised patients, acute HEV infection can develop into chronic infection, characterized by rapid progression from liver fibrosis to cirrhosis [7].

In patients infected with the human immunodeficiency virus (HIV), the course of HEV infection may involve additional problems to those found in the general population. In immunosuppressed HIV-infected patients, acute HEV infection can progress to a chronic form [7]. Furthermore, HIV patients are often co-infected with hepatotropic viruses (B and C), in which case, the course of acute HEV infection can be more severe and lead to progression or decompensation of chronic liver disease [6]. The question of whether people infected with HIV constitute a population with increased susceptibility to infection by HEV has also been raised. Many studies have evaluated the relationship between HEV and HIV, although the reported findings are controversial. Some studies have found that HIV-infected patients are at greater risk of infection by HEV [8,9,10,11], while others have found the opposite to be true [12,13,14,15,16]. This is a crucial point in the sense that determining whether HIV-infected patients indeed constitute a population at risk of HEV infection will influence the decision to apply prevention and control measures. Therefore, the aim of this systematic review and meta-analysis was to evaluate whether HIV-infected patients constitute a risk group for HEV infection.

## 2. Materials and Methods

### 2.1. Search Strategies and Selection Criteria

This analysis is reported in line with the preferred reporting items for systematic reviews and meta-analyses (PRISMA) [17] (Appendix A). A systematic literature search of the PubMed database was performed, using a combination of the terms “HEV” and “HIV”, for articles on HEV associated with HIV infection published between January 1990 and August 2018. Two independent researchers conducted the search and, based on title and abstract, determined whether the studies were related to the objective of our study (Figure 1).

### 2.2. Selection of Studies

The full texts of the selected articles were reviewed. The criteria for inclusion of the studies were as follows: i) Articles written in English or Spanish; ii) studies evaluating anti-HEV IgG by ELISA in HIV-infected patients; iii) studies comparing HEV seroprevalence between HIV patients and a control group; and iv) studies with control group with at least 50 patients included, matched by sex, age and geographical area. In addition, a second review was performed with the same criteria excluding matched control groups. The two researchers carried out the selection independently. Conflicts over decisions were resolved through a discussion or the involvement of a third investigator (Figure 1).

### 2.3. Data Extraction and Analysis

The selected studies were read to extract the data. The data of interest extracted were: Author, year of publication, country of sampling, definition of the general population, sample size, the number of individuals uninfected and infected by HEV, number of individuals uninfected and infected by HIV, and type of ELISA assay employed. The odds ratio (OR) was calculated to determine the risk of becoming infected by HEV in each group; the data were calculated if they were not expressed in the article. The data were collected by one researcher and confirmed by another. The number of infections were calculated in studies showing only sample size and prevalence of HEV. The data was collected systematically and using a standardized approach.

### 2.4. Assessment Quality of Studies

We used the Newcastle-Ottawa Scale (NOS) for assessing the quality of nonrandomized studies in meta-analyses [18]. This scale uses a ‘star system’ which judges three domains with a maximum of 9 points: The selection of the study groups (4 points); the comparability of the groups (1 point); and the ascertainment of either the exposure or outcome of interest for case-control or cohort studies respectively (3 points). A study can be awarded a maximum of one star for each numbered item within the selection and exposure categories. A maximum of two stars can be given for comparability.

### 2.5. Statistical Analysis

To compare HEV infection between HIV-infected individuals and the control groups, the ORs were calculated and 95% confidence intervals (CI) estimated, as well as *p* values. Heterogeneity between studies was calculated using the I^2^ statistic: I^2^ > 75% was high heterogeneity and <25% was low heterogeneity. To obtain the summary measure, the fixed effect model was used with the inverse-variance weighting method, which considers only intra-study variability, and the random effects model, which also assesses the existence of variability of results between the different studies. Possible publication bias was evaluated using the funnel plot method. The review manager (RevMan) (Computer program), version 5.3. Copenhagen: The Nordic Cochrane Centre, The Cochrane Collaboration, 2014, was used for data analysis.

## 3. Results

### 3.1. Data Recovery and Study Selection

In the literature search, a total of 195 citations were found in the PubMed database, from which 53 studies were selected that potentially related to the objective of our study. Fifty were subsequently excluded after reading the full text because they failed to meet all the inclusion criteria. One article was excluded due to criterion i, 24 due to criterion iii, and 25 articles due to criterion iv (Appendix A). Three papers were finally selected according to the quality evaluated by NOS, which is shown in Table 1. Subsequently, the data were extracted to perform the meta-analysis. The first study conducted by Abravanel et al. in France included 900 individuals in a 1:2 ratio of HIV-positive to healthy donors (300 HIV-positive and 600 blood donors). Of the HIV-infected patients, 116 (38.7%) were positive for anti-HEV IgG, and 284 (41.3%) were healthy donors [19]. The second study conducted by Boon et al. in Uganda included a total of 985 individuals: 491 not infected with HIV, identified from a prospective population-based cohort, and 494 HIV-infected. In both populations, a high prevalence of anti-HEV IgG antibodies was found: 46.4% in HIV patients and 47.7% in the control group [20]. Finally, the third study conducted by Bura et al. in Poland included 490 individuals. Of these, 244 were HIV-infected and 246 were blood donors. In the HIV and control groups, a total of 124 (50.8%) and 122 (49.6%) subjects, respectively, were positive for anti-HEV IgG [21]. Full details of these studies can be found in Table 1.

In the second review, 17 articles out of the 53 potentially related to the objective of our study were selected. Thirty six articles were excluded after reading the full text because they failed to meet all the inclusion criteria. One article was excluded due to criterion i, 24 due to criterion iii, and 11 articles due to criterion iv (Appendix A).

### 3.2. Meta-Analysis

In the first analysis, a total of 2375 individuals were included in this meta-analysis: 1038 of these were HIV-positive patients (43.7%) and 1337 were healthy donors (56.3%). Of the 1038 HIV patients included, 469 were positive for anti-HEV IgG antibodies (45.2%) and 569 were negative for anti-HEV IgG (54.8%). In the population of healthy donors, a total of 640 subjects were HEV anti-IgG positive (47.9%) and 697 were negative (52.1%) (Table 1).

The overall OR calculated for HIV patients in the meta-analysis was 0.87 (95% CI: 0.74–1.03) for fixed effects (Figure 2A), and 0.88 (95% CI: 0.70–1.11) for random effects (Figure 2B), with a *p* value of 0.11 and 0.27, respectively. Heterogeneity was identified as moderate according to the chi-squared test (X^2^ = 3.78), the I^2^ statistic (I^2^ = 47%) and the visual representation of the forest plots for both the fixed-effects and random-effects analyses (Figure 2). In the random-effects analysis, the Tau^2^ statistic was 0.02. The funnel plot showed absence of publication bias (Figure 3), with the point cloud distributed symmetrically around the summary measure of the effect.

In the second meta-analysis, the OR calculated was 1.07 (95% CI: 0.71–1.61) for random effects with a *p* value of 0.75. Heterogeneity was identified as severe according to the chi-squared test (X^2^ = 176.01), the I^2^ statistic (I^2^ = 91%) and the visual representation of the forest plots for random-effects (Figure 4). The funnel plot did not show absence of publication bias (Figure 5), with the point cloud distributed asymmetrically around the summary measure of the effect.

## 4. Discussion

The results of our meta-analysis do not show that HIV-infected patients are at an increased risk for HEV infection compared with healthy subjects.

We selected comparative studies in the general population matched by sex, age and area. The reason for eliminating studies that were not sex-matched was that males are at a higher risk of HEV infection than females [22,23]. Similarly, age-matched studies were a requirement because some studies have found age to be a risk factor for HEV infection [10,14,24,25], with the population aged 50 years or more being more susceptible to HEV infection. Finally, it was obligatory for studies to be matched by geographical area because this variable has been identified as an important risk factor for HEV infection [26,27,28], and because the genotypes and transmission routes of HEV vary according to geographical area. Studies that did not control for these risk factors could have caused the interpretation of results to be biased and so were excluded from the meta-analysis.

Our study selection controlled for all these aspects, which is why the final number of studies included in the meta-analysis is low, with only three articles. The meta-analysis included fixed and random effects because the level of heterogeneity was identified as moderate, with no significant differences between the two analyses. It can therefore be interpreted that heterogeneity did not influence our meta-analysis and that there was neither intra-study variability nor variability of results between the different studies. The ORs in both analyses, as well as in the three studies separately, were similar and did not identify HIV as a risk factor to be infected by HEV. Additionally, in the second systematic review and the meta-analysis—in which more articles were included because we used less restrictive criteria—it was observed that being HIV positive was not a risk factor for HEV infection with an OR similar to the previous meta-analysis. Nevertheless, more studies evaluating the relationship between possible susceptibility in HIV-infected patients and the various HEV genotypes are necessary.

As a possible limitation to the interpretation of the findings of both meta-analyses, we should consider that not all included studies report seroprevalences according to CD4^+^ cell count. Thus, the immune status of HIV infected patients compared with control is unknown. In this sense, it is known that a low count of CD4 + cells may delay or lack IgG seroconversion. Furthermore, patients with positivity to IgG anti-HEV, could show seroreversion due to a reduction of CD4+ count [23]. However, we could not control this variable in our study. Consequently, the seroprevalence reported in HIV infected patients could be underestimated due to an unknown proportion of false negative individuals.

## 5. Conclusions

The present study does not show that HIV-infected patients are at a higher risk of being infected by HEV compared with HIV-negative individuals. For this reason, despite the possible limitations, there are no arguments that recommend taking additional preventive measures for patients infected with HIV compared with those recommended for the general population.

## Figures and Tables

**Figure 1 microorganisms-07-00618-f001:**
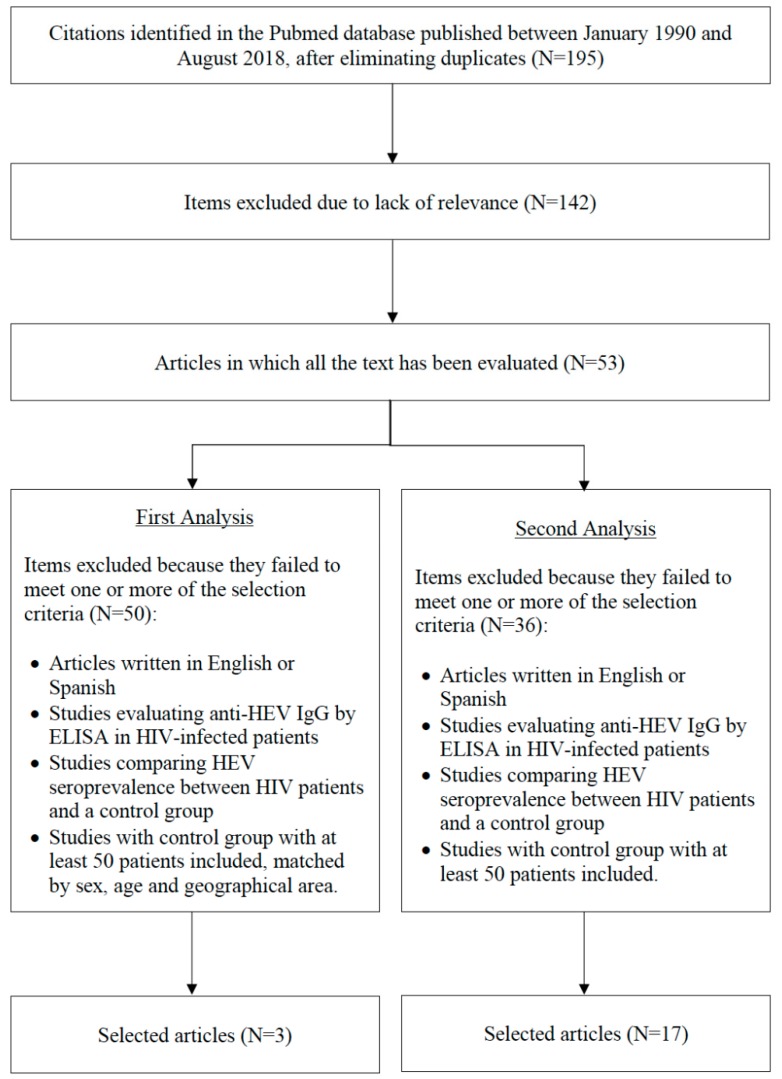
Flow diagram for the selection of studies.

**Figure 2 microorganisms-07-00618-f002:**
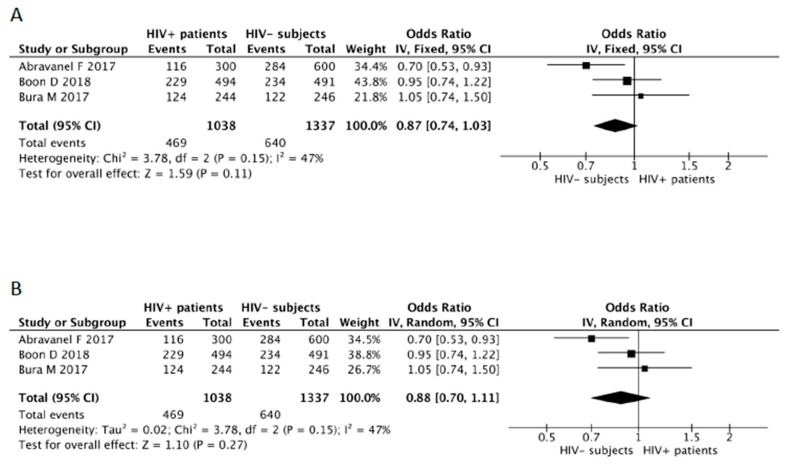
First meta-analysis of the HIV-associated risk of being infected by HEV. Lines represent the OR in meta-analysis and 95% CI, estimated using the inverse variance method in the fixed effects model (**A**), and the estimate of heterogeneity (I^2^, P het) in the random effects model (**B**).

**Figure 3 microorganisms-07-00618-f003:**
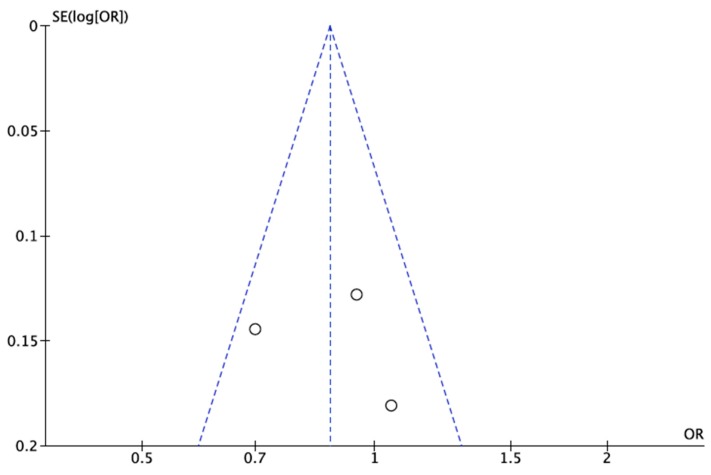
Representation of publication bias by funnel plot of the first meta-analysis. The point cloud is distributed symmetrically around the summary estimate of the effect, indicative of absence of bias.

**Figure 4 microorganisms-07-00618-f004:**
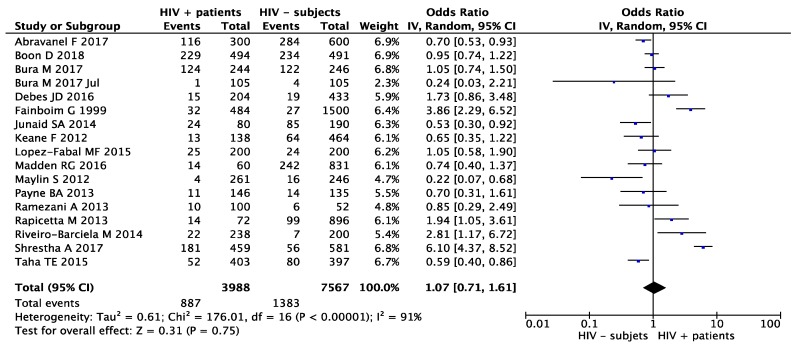
Second meta-analysis of the HIV-associated risk of being infected by HEV. Lines represent the OR in meta-analysis and 95% CI and the estimate of heterogeneity (I^2^, P het) in the random effects model.

**Figure 5 microorganisms-07-00618-f005:**
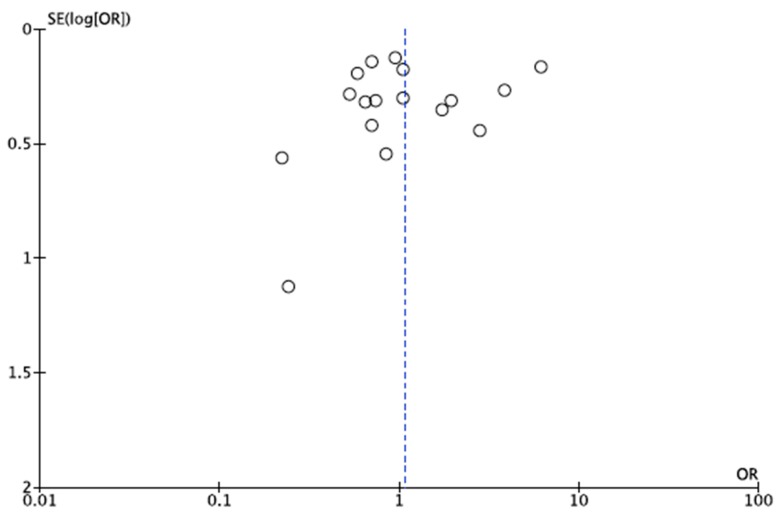
Graphical representation of publication bias by funnel plot of the second meta-analysis. The point cloud is distributed symmetrically around the summary estimate of the effect, and indicates the possible presence of bias.

**Table 1 microorganisms-07-00618-t001:** List of included studies in the first meta-analysis, evaluation of quality by NOS, and data extracted for the meta-analysis.

Included Studies				NOS for Assessing the Quality of Studies*				Extracted Data							
Author	Year	Place of Study	Tittle	Selection	Comparability	Exposure	Total Score	HIV-Positive Patients			HIV-Negative Subjects			OR (95% CI)	*p* Value
								Total Patients Included	HEV + Patients	%	Total Subjects Included	HEV + Patients	%		
Abravanel et al. [19]	2017	France (Europe)	HEV infection in French HIV-infected patients	****	**	**	8	300	116	38.7	600	284	41.3	0.70 (0.53–0.93)	0.0138
Boon et al. [20]	2018	Uganda (Africa)	Hepatitis E Virus Seroprevalence and Correlates of Anti-HEV IgG Antibodies in the Rakai District, Uganda	****	**	**	8	494	229	46.4	491	234	47.7	0.95 (0.74–1.22)	0.6824
Bura et al. [21]	2017	Poland (Europe)	Hepatitis E virus IgG seroprevalence in HIV patients and blood donors, west-central Poland	****	**	**	8	244	124	50.8	246	122	49.6	1.05 (0.74–1.50)	0.7861
TOTAL								1038	469	45.2	1337	640	47.9	-	-

Human immunodeficiency virus (HIV); Hepatitis E virus (HEV); The Newcastle-Ottawa Scale (NOS); odds ratio (OR); 95% confidence interval (95% CI). * The “star system” can be found on the website: http://www.ohri.ca/programs/clinical_epidemiology/oxford.asp.

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
