# Peer review of "Human Immunodeficiency Virus Infected Patients are Not at Higher Risk for Hepatitis E Virus Infection: A Systematic Review and Meta-Analysis"

_microorganisms, 2019, doi:10.3390/microorganisms7120618_

Round 1
Reviewer 1 Report
The authors have conducted a very useful meta-analysis of studies investigating the susceptibility of HIV infected individuals to HEV infection, answering an important research question in the field of HEV research. I have some minor comments only.
In the introduction section, line 27, it is stated that genotypes 1-4, 7 and 8 cause infection in humans. However, I am not sure the ability of genotype 8 to infect humans has been demonstrated in vivo?
In section 2.2, line 60, the text seems to contradict figure 1 as it states that in the second review 'matched by age and sex' was excluded. However, it appears in figure 1 that this is included and that only 'matched by geographical location' is excluded. Could the authors clarify this please?
Lastly, it would be good if the resolution of figure 4 could be improved.
Thank you
Author Response
Q1: In the introduction section, line 27, it is stated that genotypes 1-4, 7 and 8 cause infection in humans. However, I am not sure the ability of genotype 8 to infect humans has been demonstrated in vivo?
R1:We have changed the sentence in the way the reviewer recommends, for which we have added a new reference. In text “Five major HEV genotypes (1-4 and 7) cause infection in human [2], in addition, it has been experimentally demonstrated that genotype 8 can infect nonhuman primates, suggesting a potential zoonotic transmission [3]”.
Q2: In section 2.2, line 60, the text seems to contradict figure 1 as it states that in the second review 'matched by age and sex' was excluded. However, it appears in figure 1 that this is included and that only 'matched by geographical location' is excluded. Could the authors clarify this please?
R2: Effectively, this is a mistake. Studies included in the second analysis did not required to have a matched control group, just a control group.
Q3: Lastly, it would be good if the resolution of figure 4 could be improved.
R3: We have increased the quality of Figure 4.
Reviewer 2 Report
This meta-analysis has been well performed. The analysis is in line with the PRISMA recommendations. Without any doubt the analysis and the presented data are correct. However, the interpretation of the findings that no difference regarding the risk of HEV exposure could be observed between HIV patients and controls has some relevant limitations: -perhaps some HIV infected patients had false negative results due to an impairment of their immune response. This should be discussed in more detail.
Author Response
We would like to thanks reviewer comments and suggestions.
Q1: This meta-analysis has been well performed. The analysis is in line with the PRISMA recommendations. Without any doubt the analysis and the presented data are correct. However, the interpretation of the findings that no difference regarding the risk of HEV exposure could be observed between HIV patients and controls has some relevant limitations: -perhaps some HIV infected patients had false negative results due to an impairment of their immune response. This should be discussed in more detail.
R1: Now, we have discussed this point in more detail in a paragraph of the discussion following reviewer suggestion. We have included: "As a possible limitation to interpret the findings of both meta-analyzes, we should consider that not all included studies report seroprevalences according to CD4+ cells count. Thus, the immune status of HIV infected patients compared with control is unknown. In this sense, it is known that a low count of CD4 + cells may delay or lack IgG seroconversion. Furthermore, patients with positivity to IgG anti-HEV, could show seroreversion due to a reduction of CD4+ count [29]. However, we cannot control this variable in our study. Consequently, the seroprevalence reported in HIV infected patients could be underestimated due to an unknown proportion of false negative individuals".